adolescents; depressive symptoms; prevalence; associated factors; Vietnam

**Corresponding author:**
Cheerawit Rattanapan;
Email: cheerawit.rat@mahidol.ac.th

# Prevalence of depressive symptoms and their determinants among adolescents in Can Tho City, Vietnam: A cross-sectional study

Le Cong Tru Tran[1,2] , Cheerawit Rattanapan[3], Orapin Laosee[3], Thunwadee Tachapattaworakul Suksaroj[3], Ruochen Du[4], Sheena Ramazanu[5], Thanh Nam Truong[6], Thi Bich Son To[2] and Shefaly Shorey[7]

[1]Graduate Program of Health and Sustainable Development, Mahidol University, Phutthamonthon, Salaya, Nakhom Pathom, 73170, Thailand; [2]Medical Education and Skills Training Center, Can Tho University of Medicine and Pharmacy, Can Tho, 94117, Vietnam; [3]ASEAN Institute for Health Development, Mahidol University, Phutthamonthon, Salaya, Nakhom Pathom, 73170, Thailand; [4]Biostatistics Unit, Yong Loo Lin School of Medicine, National University of Singapore, Singapore; [5]School of Nursing and Health Studies, The Jockey Club Institute of Healthcare (IOH), Hong Kong Metropolitan University, Hong Kong, China; [6]Faculty of Public Health, Can Tho University of Medicine and Pharmacy, Can Tho, 94117, Vietnam and [7]Alice Lee Centre for Nursing Studies, Yong Loo Lin School of Medicine, National University of Singapore, Singapore

## Abstract

This study aimed to determine the prevalence and associated factors of depressive symptoms among adolescents in Can Tho City, Vietnam. A cross-sectional study was conducted with 1,054 students aged 15–18 years, recruited from eight high schools using one-off anonymous questionnaires. Depressive symptoms were assessed using the Center for Epidemiologic Studies Depression Scale Revised – Vietnamese version. The Self-esteem Scale of Vietnamese Adolescents, the Crandell Cognitions Inventory-Short form scale, the School Connectedness Scale and the Educational Stress Scale for Adolescents were used to assess self-esteem, cognitive distortion, school connectedness and educational stress, respectively. Univariate analyses explored the relationships between sociodemographic variables and depressive symptoms. Pearson correlations were calculated for the associations between variables. Multiple regression was used to adjust for the factors that contributed to depressive symptoms in adolescents. The findings revealed that 37.4% of adolescents in Can Tho City, Vietnam, experienced depressive symptoms. Factors influencing depression in adolescents include cognitive distortions, academic pressure, exposure to interpersonal violence, consumption of alcohol and smoking, family history of depression, family incarceration and experiences of digital sexual violence. These results underscore the urgent need for a multilevel and multidimensional intervention strategy involving parents, educators, mental health professionals and policymakers to promote early identification, provide support and enhance mental health literacy among adolescents.

## Impact statement

This study revealed a significant prevalence of depressive symptoms among adolescents in Can Tho City and identified several critical contributing factors, such as cognitive distortions, academic pressure, exposure to domestic violence, family incarceration, alcohol consumption and smoking among adolescents, bullying and experiences of trauma from digital sexual violence. These results highlight the urgent need for comprehensive mental health interventions that address modifiable risk factors. A coordinated, multilevel strategy involving parents, mental health professionals, schools and the broader community is essential for addressing these issues through the early identification of at-risk individuals, the provision of support and mental health literacy education for adolescents and relevant stakeholders. Moreover, it is vital to equip adolescents with skills to reframe negative thoughts, cope with academic stress and safeguard themselves against interpersonal and sexual violence to reduce the prevalence of depression in this demographic. Future research should focus on exploring the factors influencing depressive symptoms among marginalized and vulnerable groups in Vietnam, including ethnic minorities, gender and sexual minorities and low-income and rural populations.

## Introduction

Depression is a significant global public health concern, particularly among adolescents, a population undergoing profound cognitive, emotional and social transitions, especially those

aged 15–18 years (World Health Organization, 2024). The World Health Organization (WHO) identifies depression as a leading contributor to disability-adjusted life years, with adolescents being especially vulnerable to its effects (The Lancet Global, 2020; WHO, 2022). Depression during adolescence is associated with academic difficulties, impaired social relationships, increased risk-taking behaviors, substance use and suicidal ideation (Ledford, 2014; Pozuelo et al., 2022). Despite growing global awareness, adolescent depression remains particularly prevalent in low- and middle-income countries (LMICs), where access to mental health resources is often constrained (Girma et al., 2021).

Vietnam, a rapidly urbanizing country, faces mounting challenges in addressing adolescent mental health needs (Vuong et al., 2011). With over 14 million adolescents, depression represents an underrecognized yet growing concern (UNICEF, 2023b). Existing research has reported varying rates of adolescent depression in Vietnam, ranging from 22.9% (Thai et al., 2020) to 50% (Lê et al., 2023). However, depression-related services remain underdeveloped, with a severe shortage of trained professionals and limited integration of psychological care within primary healthcare settings in school climates (WHO, 2005). The cultural stigma surrounding depressive disorders further impedes early detection and intervention, exacerbating the burden of untreated adolescent depression (Do et al., 2014).

Despite prior studies highlighting the prevalence of adolescent depression in Vietnam, there is a dearth of comprehensive, up-to-date research focusing on the specific determinants of depressive symptoms across diverse sociocultural and economic contexts. Can Tho City, the largest city in the Mekong Delta region of Vietnam, is considered a rural and semi-urban area, yet it has seen rapid urbanization, economic growth and internal migration. It is also one of the five municipalities under the direct control of the central government, allowing for direct access to national policies and funding (Dao, 2024). The last comprehensive data on adolescent depression in Can Tho City was collected in December 2011, revealing that 41.1% of adolescents experienced depressive symptoms (Nguyen et al., 2013). Moreover, the current generation's increased exposure to digital technology has been linked to issues such as gaming and social media addiction, decreased social connectedness, physical inactivity and reduced sleep quality (Dam et al., 2023; Ho, 2019; Owens et al., 2016).

Adolescent depressive symptoms result from a complex interaction of multilevel determinants. Previous studies have consistently demonstrated the influence of individual-level determinants, such as age (Wartberg et al., 2018), sex (Salk et al., 2016), body image (Murray et al., 2018), chronic conditions (Pinquart & Shen, 2011), health-risk behaviors (Audrain-McGovern et al., 2009), inactive activity (Costa et al., 2021), family history of depression (Kwong et al., 2019), serious injuries (Jacob et al., 2020), self-esteem (Orth et al., 2014), gaming online (Hellström et al., 2015), academic pressure (Jayanthi et al., 2015) and negative thinking (Marton et al., 1993) on depressive symptoms in adolescents. Beyond personal determinants, interpersonal-level determinants, including family economic levels (Reiss, 2013), family environment (Lau & Kwok, 2000), school environment (Brière et al., 2013), school connectedness (Raniti et al., 2022) and loss of a loved one (grief) (Kaplow et al., 2021), have also been identified as critical contributors to adolescent depressive symptoms. More recently, the influence of community-level determinants, such as experiences of sexual violence (Meadows et al., 2022), cyberbullying (Hu et al., 2021) and social media use (Ivie et al., 2020), has been highlighted as emerging risk factors for the existing young generation.

Our study adopts a social-ecological model to conceptualize the multifaceted influences on adolescent depression. This framework posits that mental health outcomes are shaped by interactions across multiple levels: individual, interpersonal and community levels. This approach aligns with the recent study emphasizing the role of neighborhood and environmental factors in adolescent mental health, particularly in LMICs. It recognizes that addressing adolescent depression requires a comprehensive understanding of individual vulnerabilities within broader social and structural contexts (Roy & Sowgat, 2024). However, the interaction between these multilevel factors remains underexplored in the Can Tho City context. By addressing these gaps, this study will provide an updated and contextually relevant understanding of adolescent depression, thereby informing evidence-based interventions.

This study aims to assess the prevalence and associated factors of depressive symptoms among adolescents in Can Tho City, Vietnam. By contributing new insights into the determinants of adolescent depression, this study supports ongoing efforts to enhance mental health services and policies in Vietnam. Its findings may also offer valuable perspectives for other LMICs confronting similar adolescent mental health challenges.

## Methodology

### Study design

A cross-sectional study was conducted to assess the prevalence and factors associated with depressive symptoms among adolescents aged 15–18 years in Can Tho City, Vietnam, from June to August 2024.

### Participants

Adolescents were eligible for the study if they resided in Can Tho city, were enrolled in grades 10 and 11 at the selected high schools and had obtained consent from their parents or legal guardians to participate in the study. Exclusion was applied to students who did not assent, were unable to obtain consent from their parents or legal guardians or had physical and/or mental conditions that interfered with their ability to complete the survey.

This study employed a multistage sampling technique. In the first stage, a single high school was randomly selected from each district, resulting in the selection of nine high schools from nine districts. Subsequently, two classes from grades 10 and 11 were randomly selected from each high school, totaling 36 classes and ~1,440 adolescents. One of the nine selected high schools declined to participate, leaving eight high schools in the study. Out of 1,280 eligible students, 174 students declined to participate in the study, while 1,106 students consented and participated. After excluding incomplete questionnaires, 1,054 questionnaires were deemed eligible for data analysis.

### Sample size

Based on an estimated design effect of 1.5–2 and a margin of 5%, the sample size was calculated to be 744 with power at 80% and significance level at 5% (Hulland et al., 2016; Islam et al., 2021). To account for a 20% dropout rate, the final minimum sample size was adjusted to 930 adolescents.

## Procedure

After obtaining ethics approval from Mahidol University (Certificate of Approval No. MU-CIRB 2024/153.0606), Can Tho University of Medicine and Pharmacy (No. 24.001/PCT-HDDD) and the Can Tho Department of Education and Training (No. 1518/SGDDT-CTTT) for collecting data, nine school principals were contacted to seek permission to collect data from the respective nine selected schools. As one school declined to participate, the study was conducted in eight schools. After soliciting permission, four trained research assistants (RAs) visited the schools (one RA to two schools ratio) to recruit and collect data. The RAs screened eligible adolescents for the study and explained the study details to them in their schools. After the adolescents had given their written assent and their parents had concurrently given consent, the adolescents were invited to a quiet room within the school to complete the paper-based questionnaires. Throughout the data collection period, the research team addressed any participant concerns regarding the questionnaire content. To ensure anonymity, return boxes with covers for dropping off the questionnaires were available inside the room and were sealed after all participants had submitted their questionnaires. Additionally, all participants received a small box containing various stationery items (equivalent to 1.60 US dollars of remuneration) after completing the questionnaires.

## Measurement

Data were collected using the authors' developed questionnaires combined with the UNICEF survey (Unicef, 2023a), the Global School-based Student Health Survey core-expanded Questions 2021 version (WHO, 2021) and the Adverse Childhood Experiences International Questionnaire (WHO, 2020), collecting data on adolescents' demographics, family and social lives. Validated tools, such as the Center for Epidemiologic Studies Depression Scale Revised–Vietnamese Version (CESDR-V) (Tran et al., 2022), the Self-esteem Scale of Vietnamese Adolescents (EVES) (Linh et al., 2017), the Crandell Cognitions Inventory-Short form (CCI-SF) scale (Dekker et al., 2011), the School Connectedness Scale (SCS) (Bonny et al., 2000; Ozer, 2005) and the Educational Stress Scale for Adolescents (ESSA) (Sun et al., 2011), as detailed below, were also used.

The CESDR-V consists of 20 items that assess depressive symptoms experienced in the past 2 weeks. Each item in the questionnaire is rated on a 5-point Likert scale, ranging from 0 (equivalent to "*Not at all or <1 day in the past week*") to 4 (equivalent to "*Nearly every day for 2 weeks*"). The minimum score for this instrument is 0, and the maximum score is 80; the higher the score, the higher the depressive symptoms (Tran et al., 2022). The score was classified into no depressive symptoms (<16 points), at risk for depression (16–21 points), elevated level of depressive symptoms (22–25 points) and level of depressive symptoms comparable with major depressive disorder (>25 points) (Nguyen et al., 2013). This scale was validated for Vietnamese adolescents with high reliability (Cronbach's $\alpha = 0.92$) and validity (Tran et al., 2022). The reliability of this scale in this study remained high (Cronbach's $\alpha = 0.92$).

The EVES measures five dimensions of self-esteem: the academic self, physical self, emotional self, future self and social self (Oubrayrie et al., 1996; Sordes-Ader et al., 1998). The scale, adapted from the original European Self-Esteem Scale of Toulouse to Vietnamese local culture, contains 33 items that are rated using a 5-point Likert scale; with 0 being equal to "S*trongly disagree*" and 4 being equal to "S*trongly agree*" (Linh et al., 2017). The minimum

score is 0, and the maximum score is 132; the higher the score, the higher the self-esteem. This scale was validated for Vietnamese adolescents with high reliability (Cronbach's $\alpha = 0.86$) and high test–retest reliability (Intraclass Correlation Coefficient (ICC) = 0.91, 95% confidence interval [CI] = 0.84, 0.95, $p < 0.001$) (Linh et al., 2017). The Cronbach's alpha of this scale used in this study was acceptable at 0.615.

The CCI-SF scale measures cognitive distortions or the frequency of depressive thoughts. It includes 12 items with a 5-point Likert scale. The minimum score is 12, and the maximum score is 60. The CCI-SF was translated into Vietnamese by two invited mental health professionals. Thereafter, a panel of five experts examined the CCI-SF's content validity. Each item's item-objective congruence (IOC) was calculated, and the overall test IOC was acceptable (0.963). The short-form scale was validated among patients in America with heart failure and demonstrated acceptable reliability (Cronbach's $\alpha = 0.84$) and construct validity ($r = 0.92$, $p < 0.001$) (Dekker et al., 2011). In this study, the internal consistency of the scale was very high (Cronbach's $\alpha = 0.916$).

The SCS was first developed based on the data from the National Longitudinal Study of Adolescent Health (Add Health) to measure adolescents' sense of connectedness to their school, characterized by their perceived closeness with the school personnel and school environment (Resnick et al., 1997). The scale was originally developed based on six items, but was revised to five items rated on a 5-point Likert scale in subsequent studies (Bonny et al., 2000; Ozer, 2005). Each item is evaluated with 0 = *strongly disagree*, 1 = *disagree*, 2 = *neither disagree nor agree*, 3 = *agree* and 4 = *strongly agree*. The cutoff scores are 10 and 16, with three levels, including weak connectedness (<10 points), average connectedness (10–16 points) and strong connectedness (>16 points) (Ho, 2019). This scale has been validated among 18 different sociocultural groups in North America and reported high reliability (Cronbach's $\alpha = 0.82$–$0.88$) and concurrent validity ($r = 0.44$–$0.55$) (Furlong et al., 2011). Furthermore, this scale has been widely used across countries with acceptable reliability, including Vietnamese adolescents (Ho, 2019; Law et al., 2022). The internal consistency of this scale used in this study is high (Cronbach's $\alpha = 0.791$).

The ESSA measures the academic stress of adolescents, consisting of 16 items rated on a 5-point Likert scale ranging from 1 (*strongly disagree*) to 5 (*strongly agree*) (Sun et al., 2011). The minimum score is 16, and the maximum score is 80. The higher the scores, the higher the academic pressure. The cutoff score used in the current study comprises three levels, including low stress (<50), medium stress (51–58) and high stress (>58) (Nguyen et al., 2013). It was validated and used for Vietnamese adolescents with high reliability (Cronbach's $\alpha = 0.83$ to $0.86$) and validity (Ho, 2019; Ho et al., 2022; Truc et al., 2015). The internal consistency of the scale used in this study was very high (Cronbach's $\alpha = 0.9$).

## Data analysis

Data were analyzed using the Statistical Package for the Social Sciences version 29 (IBM Corp, 2023). In the current study, the proportion of missing data for all variables was <5%. Preliminary checks comparing participants with and without missing values, demographics and study variables demonstrated no statistically significant differences, suggesting that the missing values were unrelated to either the observed or unobserved data. This result is consistent with the assumption of missing completely at random (MCAR) (Little & Rubin, 2019). The small proportion of missing data and the MCAR indicated that using a complete case analysis

was an appropriate and valid approach that would not introduce crucial bias into the results. Moreover, weighting was not applied in the analysis because the sampling strategy was designed to ensure that the distribution of participants was already proportionally representative of the target population. A stratified random sampling approach was employed in which the number of participants selected from each high school and grade level was proportional to their actual distribution in the total target population of Can Tho City. Additionally, the number of classes and the number of students in each class are equivalent, as stipulated by the Ministry of Education and Training of Vietnam. This proportion allocation minimized sampling bias and ensured that each participant had an approximately equal probability of selection. As a result, the sample closely reflected the demographic structure of the population, making additional weighting unnecessary.

Descriptive statistics, including frequencies, percentages (%), means (M) and standard deviation (SD), were employed to summarize participant characteristics. Based on the CESDR-V scores, participants were categorized into four groups: no depressive symptoms, at-risk for depression, elevated level of depressive symptoms and level of depressive symptoms comparable with major depressive disorder. One-way analysis of variance (ANOVA) and Student's *t*-tests were used to examine the association between categorical variables and depressive symptoms in adolescents. Pearson correlations between depressive symptoms and other continuous variables were calculated. A total of 29 variables with statistically significant associations with depressive symptoms in univariate analyses were then selected to be entered into the multiple linear regression model. A multiple linear regression (stepwise elimination) was used to adjust for the factors that contributed to depressive symptoms in adolescents. All statistical tests were two-tailed, and a *p*-value $< 0.05$ was considered statistically significant.

## Results

### *The demographic characteristics of the participants*

Of the 1,054 adolescents who completed the questionnaires, 48.8% were males ($n = 514$), and 51.2% were females ($n = 540$). Half of the participants ($n = 612$, 58.1%) resided in urban areas, while the other half resided in rural areas ($n = 442$, 41.9%). Most of the participants did not report any chronic physical illnesses ($n = 888$, 84.3%), nor a family history of depression ($n = 888$, 84.3%). The mean age of participants was 16.4 (SD = 0.616), and the mean Tri-Ponderal Mass Index (TMI) was 12.3 (SD = 2.26). Many of the adolescents ($n = 1,011$, 95.9%) lived on a monthly household income per capita of at least 60.95 USD (if residing in rural areas) or 81.30 USD (if residing in urban areas). More details are provided in Table 1.

### *Prevalence of depressive symptoms*

The mean depressive symptoms score was 15.0 (SD = 13.8). The results indicated that 62.6% of the adolescents did not exhibit depressive symptoms (CESDR-V $< 16$), while 37.4% showed depressive symptoms (CESDR-V $\geq 16$). Table 2 illustrates the prevalence of depressive symptom levels among the participants.

### *Individual, family and social factors associated with depressive symptoms*

Independent-sample *t*-tests and one-way ANOVA were used to examine the associations between individual, family and social

**Table 1.** The demographic of the participants ($n = 1,054$)

| Variables | Mean (SD) | *n* | % |
|---|---|---|---|
| **Gender** | | | |
| Male | | 514 | 48.8 |
| Female | | 540 | 51.2 |
| **Age** | 16.4 (0.616) | | |
| **Geographical living area** | | | |
| Urban | | 612 | 58.1 |
| Rural | | 442 | 41.9 |
| **Chronic physical conditions** | | | |
| Yes | | 40 | 3.8 |
| No | | 888 | 84.3 |
| Do not know | | 126 | 12 |
| **TMI (Tri-Ponderal Mass Index)** | 12.3 (2.26) | | |
| Normal | | 999 | 94.8 |
| Overweight | | 45 | 4.3 |
| Obese | | 10 | 0.9 |
| **Family history of depression** | | | |
| Yes | | 61 | 5.8 |
| No | | 888 | 84.3 |
| Do not know | | 105 | 10.0 |
| **Monthly household income per capita** | | | |
| <60.95 USD (rural areas) or <81.30 USD (urban areas) | | 43 | 4.1 |
| 60.95–91.43 USD (rural areas) or 81.30–121.90 USD (urban areas) | | 500 | 47.4 |
| >91.43 USD (rural areas) or >121.90 USD (urban areas) | | 511 | 48.5 |
| **Days of physical activity in a week** | 3.93 (2.31) | | |
| **Experience of school bullying** | | | |
| Yes | | 98 | 9.3 |
| No | | 956 | 90.7 |
| **Experience of cyberbullying** | | | |
| Yes | | 86 | 8.2 |
| No | | 968 | 91.8 |
| **Total number of hours per school day using social media** | | | |
| <1 h | | 50 | 4.7 |
| 1–2 h | | 137 | 13.0 |
| 3–4 h | | 281 | 26.7 |
| 5–6 h | | 247 | 23.4 |
| 7–8 h | | 128 | 12.1 |
| >8 h | | 211 | 20.0 |

factors with adolescents' depressive symptom scores. The results are presented in Supplementary Table S1. Female adolescents were at significantly greater risk for depressive symptoms ($-4.37$, 95% CI = $-6.02$ to $-2.72$) as compared to their male counterparts. Adolescents who suffer from chronic physical conditions were also significantly more likely to have depressive symptoms ($-5.01$, 95% CI = $-10.2$ to

**Table 2.** Prevalence of depressive symptom levels among participants (*n* = 1,054)

| CESDR-V* | *n* | % |
|---|---|---|
| No depressive symptoms (CESDR-V < 16 points) | 660 | 62.6 |
| At risk for depression (CESDR-V 16–21 points) | 136 | 12.9 |
| Elevated level of depressive symptoms (CESDR-V 22–25 points) | 61 | 5.8 |
| Level of depressive symptoms comparable with major depressive disorder (CESDR-V > 25 points) | 197 | 18.7 |

*CESDR-V, simplified Vietnamese version of the Center for Epidemiologic Studies Depression Scale Revised.

0.14). Consuming alcohol (−6.28, 95% CI = −8.46 to −4.09) and smoking cigarettes (−14.9, 95% CI = −21.9 to −8.03) significantly increased adolescents' risk for depression. A family history of depression greatly increased the risk for depressive symptoms in adolescents (−10.36, 95% CI = −14.4 to −6.31).

Adolescents who have experienced serious injury (−3.43, 95% CI = −5.29 to −1.56) or have lost a loved one (2.26, 95% CI = 0.60–3.92) were significantly more likely to report depressive symptoms. Victims of sexual violence (7.18, 95% CI = 2.05–12.3), intimate partner violence (−10.8, 95% CI = −19.9 to −1.91), technology-facilitated sexual violence (TFSV) (−4.14, 95% CI = −5.89 to −2.38) and forced sex (−7.05, 95% CI = −12.1 to −1.97) were significantly predisposed to higher rates of depressive symptoms. Victims of school bullying (−7.34, 95% CI = −10.6 to −4.07) and cyberbullying (−6.98, 95% CI = −10.7 to −3.29) were also significantly at greater risk for depressive symptoms.

Adolescents from middle-income households were more likely to report higher depressive symptoms as compared to high-income households (−2.23, 95% CI = −4.26 to −0.20). There was no significant difference in depressive symptom score between adolescents from low-income and high-income households. Victims of physical abuse at home by beating (8.99, 95% CI = 6.17–11.8) or use of a weapon (10.2, 95% CI = 6.24–14.2), and emotional abuse by verbal insults (8.95, 95% CI = 6.73–11.2) or neglect (10.6, 95% CI = 7.88–13.4) were at significantly greater risk for depression. Exposure to violence at home, including verbal insults (8.80, 95% CI = 6.96–10.6), beating (9.20, 95% CI = 6.70–11.7) or use of a weapon (8.04, 95% CI = 4.92–11.2), was also a significant risk factor for depression. Depressive symptoms were significantly more likely to

occur in adolescents who are living with family members who abuse substances (3.97, 95% CI = 1.56–6.38) or have been incarcerated (6.72, 95% CI = 3.09–10.4).

Participation in sports, smoking e-cigarettes and shisha, phone ownership and family rules on phone use were not associated with depressive symptoms in adolescents.

### Individual, parental factors, internet usage and other scales correlated with depressive symptoms

Pearson correlation tests were used to assess the relationship between total depressive symptoms score and age, TMI, physical activity levels, self-esteem, cognitive distortions, school connectedness, academic pressure, perceived parental involvement and internet usage. The results were reported in Supplementary Table S2.

Adolescents who spent more days in a week doing physical activities were significantly less likely to be depressed ($\rho = -0.125$, 95% CI = −0.184 to −0.065). Age and TMI were not significantly correlated with depressive symptoms.

Distorted cognitions ($\rho = 0.679$, 95% CI = 0.645–0.710) and academic pressure ($\rho = 0.516$, 95% CI = 0.470–0.559) were positively correlated with depressive symptoms. In contrast, school connectedness symptoms ($\rho = -0.178$, 95% CI = −0.236 to −0.119) were negatively correlated with depressive symptoms. Self-esteem was not significantly correlated with depressive symptoms.

Adolescents who perceived their parents to be more involved in their homework ($\rho = -0.087$, 95% CI = −0.147 to −0.027), more understanding ($\rho = -0.297$, 95% CI = −0.351 to −0.241), more interested in their hobbies symptoms ($\rho = -0.207$, 95% CI = −0.264 to −0.148) and more likely to offer advice and guidance symptoms ($\rho = -0.281$, 95% CI = −0.336 to −0.224) were less likely to be depressed. Adolescents who perceived their parents to have higher expectations of their symptoms ($\rho = 0.092$, 95% CI = 0.032–0.152) were more likely to be depressed.

Longer durations of internet use ($\rho = 0.174$, 95% CI = 0.114–0.231), screentime ($\rho = 0.165$, 95% CI = 0.106–0.223) and online gaming ($\rho = 0.095$, 95% CI = 0.035–0.155) were also positively correlated with depressive symptoms in adolescents.

### Multiple regression model

The final regression model with 10 factors associated with depressive scores among Vietnamese adolescents is presented in Table 3.

**Table 3.** Multiple regression model of the factors associated with depression scores among adolescents (*n* = 1,054)

| Variables | β | *p*-value | 95% CI | VIF |
|---|---|---|---|---|
| CCI | 0.779 | <0.001 | 0.699, 0.859 | 1.64 |
| ESSA | 0.176 | <0.001 | 0.116, 0.236 | 1.54 |
| No family history of depression | −4.27 | <0.001 | −5.91, −2.62 | 1.11 |
| Experience of domestic physical abuse (beating) | 2.29 | 0.018 | 0.398, 4.18 | 1.28 |
| Nonconsumption of alcohol | −2.26 | 0.002 | −3.69 to −0.819 | 1.10 |
| Witnessed physical abuse at home (beating) | 2.22 | 0.005 | 0.656 to 3.77 | 1.11 |
| Living with a family member who has been to prison before | 3.49 | 0.012 | 0.767 to 6.22 | 1.19 |
| Abstinence from cigarette smoking | −6.73 | 0.003 | −11.1 to −2.37 | 1.04 |
| No experience of school bullying | −2.52 | 0.023 | −4.69 to −0.343 | 1.23 |
| No experience of technology-facilitated sexual violence | −1.32 | 0.032 | −2.53 to −0.114 | 1.07 |

*Note:* CCI, Crandell Cognitions Inventory (Short Form); ESSA, Educational Stress Scale for Adolescents.

Higher rates of cognitive distortions ($\beta$ = 0.779, $p$ < 0.001) and greater academic pressure ($\beta$ = 0.176, $p$ < 0.001) are significantly associated with higher rates of depressive symptoms. Adolescents with no family history of depression ($\beta$ = −4.27, $p$ < 0.001), abstained from consuming alcohol ($\beta$ = −2.26, $p$ = 0.002) and cigarette smoking ($\beta$ = −6.73, $p$ = 0.003) and were not victims of school bullying ($\beta$ = −2.52, $p$ = 0.023) and TFSV ($\beta$ = −1.32, $p$ = 0.032) had lower depressive symptom scores. Experiencing domestic physical abuse in the form of beatings ($\beta$ = 2.29, $p$ = 0.018), witnessing physical abuse at home in the form of beatings ($\beta$ = 2.22, $p$ = 0.005) and living with a family member who has been to prison before ($\beta$ = 3.49, $p$ = 0.012) had a significant positive effect on depressive symptom scores. All significant variables had a variance inflation factor < 2, indicating no severe multicollinearity between the variables.

School connectedness, levels of physical activity, perceived parental involvement in homework, showing understanding, interest in hobbies, providing advice, guidance and expectations and duration of internet use, screen time and online gaming were excluded from the final model. The following demographics were also not significant factors associated with depressive symptoms in adolescents: gender, chronic physical conditions, experience of serious injuries, sexual violence, intimate partner violence, forced sex, loss of a loved one, cyberbullying, domestic physical abuse in the form of cutting, domestic emotional abuse via verbal insults and neglect, household income, living with a family member who abuses substances, witnessing verbal abuse at home or physical abuse in the form of cutting at home and divorced parents (see Figure 1).

## Discussion

This study assessed the prevalence of depressive symptoms and related factors among adolescents in Can Tho City, Vietnam. Most of the adolescents included in the study came from middle to high-income households (with a monthly household income per capita of more than 60.95 USD for rural households and 81.30 USD for urban households). The results show that the prevalence of depressive symptoms among Vietnamese adolescents is 37.4%. In our study, adolescents who had more cognitive distortions, felt greater

academic pressure, experienced domestic physical abuse in the form of beatings and witnessing beatings at home and lived with a family member who had been to prison before were more likely to have higher rates of depressive symptoms. Meanwhile, the absence of a family history of depression, abstinence from alcohol consumption and smoking cigarettes, not experiencing bullying at school and technology-facilitated sexual violence were protective factors against depression in adolescents.

The results of this study reveal a moderately high prevalence of depressive symptoms (37.4%) among adolescents aged 15–18 years in Can Tho City. This figure aligns with a previous study in the same region, which reported a prevalence of 41.1% among adolescents aged 15–19 years at risk for depression (Nguyen et al., 2013), suggesting that high rates of depressive symptoms have persisted over time. These findings emphasize the urgent need for preventive interventions targeting adolescent mental health. The prevalence observed in this study is also consistent with the 32% rate reported among adolescents from Ha Noi, Dong Thap, Gia Lai and Dien Bien (United Nations International Children's Emergency Fund (UNICEF) Viet Nam, 2022). However, the current study's rate is notably higher than the 4.3% depression rate found in a national survey covering 38 provinces (Institute of Sociology, 2022). Such discrepancies in prevalence may stem from variations in population demographics, study methodologies, geographical locations, data collection periods, screening tools and socioeconomic factors. For instance, the lower rate in the national survey could be attributed to the use of face-to-face interviews, where adolescents may have been hesitant to disclose depressive symptoms due to stigma surrounding mental health and low mental health literacy in the community.

Similar prevalence rates have been reported in other Southeast Asian countries, such as 26.4% in Thailand (Angsukiattitavorn et al., 2020), 19% in Malaysia (Singh et al., 2023), 27.86% in Indonesia (Purborini et al., 2021), 27.2% in Myanmar (Carroll et al., 2021), 24% in Laos (Phanthavong et al., 2013) and 8.9% in the Philippines (Puyat et al., 2021). The relatively similar rates across these countries may be attributed to their shared status as LMICs, undergoing rapid development and rural-to-urban migration, leading to changes in population structures and increased stressors on mental health. The lower prevalence in the Philippines

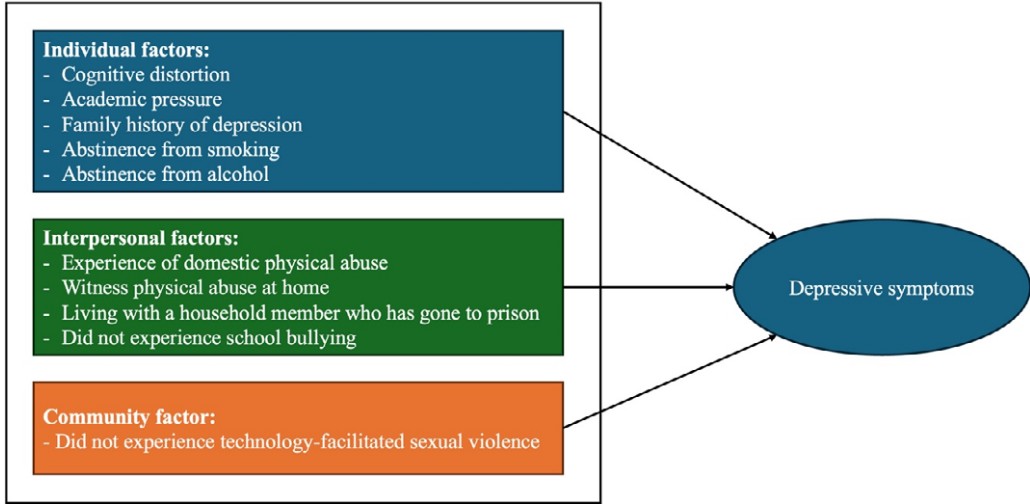

**Figure 1.** Social-ecological model of factors associated with depressive symptoms among adolescents in Can Tho City, Vietnam.

could be due to the use of higher cutoff points for moderate and severe depressive symptoms, which contrasts with the thresholds used in other studies.

This study found that higher levels of cognitive distortions were associated with increased depressive symptoms among Vietnamese adolescents. According to Beck's cognitive distortion theory, mal-adaptive thought patterns – such as cognitive rigidity, dichotomous thinking and selective abstraction – can distort perceptions of reality, leading individuals to experience depressive symptoms, particularly during stressful situations (Beck, 1963; Jager-Hyman et al., 2014). Similar to the current study, adolescents in Malaysia with more distorted thought patterns were more likely to report suicidal ideation (Jaffri et al., 2021). Meta-analyses have demonstrated that cognitive behavioral therapy (CBT), which focuses on restructuring distorted cognitions, is highly effective in reducing depressive symptoms among youth, with subclinical depression risk decreasing by 63% following CBT intervention (Idsoe et al., 2019; Oud et al., 2019). More specifically, schools could implement CBT-based interventions to help students manage stress, recognize negative thoughts and replace them with healthier thinking; integrating psychoeducation on cognitive distortions into curricula fosters awareness and resilience (Bonnesen et al., 2020; Saw et al., 2019).

Academic pressure was found to be significantly associated with depressive symptoms among Vietnamese adolescents. This may be attributed to the high value placed on educational achievement in Vietnamese society, where academic success is seen as a primary pathway to economic mobility and is closely tied to family honor (Mai & Yang, 2013). Such expectations can place disproportionate pressure on students to perform well, and failure or poor academic performance may lead to feelings of inadequacy, anxiety and shame (Nguyen & Phan, 2024). Similar findings have been reported in other Asian countries with a strong emphasis on academic achievement, such as India and China, where depressed adolescents were more likely to experience severe academic pressure (Jayanthi et al., 2015; Jiang et al., 2021). The shift toward online education following the coronavirus disease 2019 (COVID-19) pandemic has further exacerbated academic stress in adolescents across European countries, such as Turkey, Italy and Slovenia, due to challenges with internet access, difficulties with online learning modalities and reduced student motivation and concentration (Cofini et al., 2022; Durbas et al., 2021; Šorgo et al., 2022). Mental health professionals should raise awareness among adolescents and their parents regarding the psychological toll of excessive academic pressure and encourage those experiencing debilitating stress to seek professional psychological support. School-based mindfulness or self-compassion interventions could also be implemented to foster academic resilience, promote positive reframing of thoughts related to school and reduce academic stress among students (Baumgartner & Schneider, 2023; Shahid & Farhan, 2022). Future longitudinal studies could explore the impact of changes in secondary and higher education on adolescent mental health and academic pressure in Vietnam following the COVID-19 pandemic.

This study found that adolescents without a family history of depression are less likely to experience depressive symptoms. Numerous studies have established that a family history of mental health disorders is a significant risk factor for the development of mental health conditions in children (Milne et al., 2009; Rasic et al., 2014; Sullivan et al., 2000). Specifically, the biological children of parents with depression face a two- to fivefold increased risk of developing major depressive disorder themselves (Fendrich et al., 1990; Hammen & Brennan, 2003; Pilowsky et al., 2006; Sullivan et al., 2000; Van Batenburg-Eddes et al., 2013). Moreover, adolescents whose parents and grandparents have a history of depression bear a compounded risk of developing the disorder (van Dijk et al., 2021). Therefore, clinicians working with adolescents should screen for a family history of depression, as it significantly elevates their risk for mental health conditions. Targeted mental health literacy campaigns aimed at this vulnerable group are critical, as early education and awareness may mitigate their risk of developing depression. Additionally, parent–child interventions based on CBT could be effective in reducing and preventing depressive symptoms in both parents and their adolescents (Garber et al., 2009). Policymakers should also consider allocating additional resources to the mental health sector in Vietnam to enhance screening, detection and treatment of depression among parents, which may in turn improve mental health outcomes in adolescents (Thapar et al., 2010).

Abstinence from smoking and alcohol consumption is associated with a reduced risk of depression in adolescents. The findings of this study align with previous research conducted in the United States and China, which reported correlations between varying levels of alcohol consumption – ranging from light to heavy drinking – and depressive symptoms in adolescents (Crane et al., 2021; Zhang et al., 2021). Additionally, past reviews have identified robust bidirectional associations between cigarette smoking and adolescent depression (Chaiton et al., 2009; Farooqui et al., 2023). The significant link between alcohol use and depression may be attributed to the pronounced neurobiological changes that occur in the brain during adolescence, which heighten vulnerability to substance dependence and susceptibility to depressive disorders (Chaiton et al., 2009). Adolescents may be inclined to initiate smoking or drinking alcohol in response to stressful situations or negative peer influences as a form of self-medication (Bozzini et al., 2020). Therefore, clinicians and mental health professionals must proactively discourage these behaviors and educate adolescents early to prevent the initiation of smoking and alcohol consumption. Furthermore, cessation interventions aimed at adolescents already dependent on these substances require a multidisciplinary approach to effectively address the underlying social stressors driving substance use. Future research should focus on exploring the motivations and reasons behind smoking and drinking behaviors among adolescents in Vietnam to inform the development of targeted interventions.

Exposure to physical and domestic violence, including nonweapon beatings, such as spanking, slapping, kicking and punching – whether the adolescent is a direct victim or merely a witness – significantly increases the risk of depression. Meta-analyses have shown that individuals exposed to physical violence at an early age are 2.5 times more likely to develop major depressive disorder (LeMoult et al., 2020). Adverse childhood experiences, such as direct physical abuse or witnessing parental violence, have enduring and detrimental effects on neurological development, leading to lifelong consequences for mental health that can extend well into adulthood (Anda et al., 2010; Covey et al., 2017). In Vietnam, corporal punishment, a form of violent discipline, is widely accepted and normalized due to deeply ingrained social and gender norms. Qualitative studies have found that many parents and teachers view corporal punishment as an expected form of discipline, influenced by patriarchal family structures, Confucianism and the intergenerational transmission of such practices (Beazley et al., 2006; Nguyen et al., 2014; Xing et al., 2017). A recent study in Cambodia revealed that younger parents in rural areas, particularly

those with multiple children, low socioeconomic status and limited education, are more likely to hold favorable views on the use of corporal punishment (Nho & Seng, 2017). These findings highlight the urgent need for educational campaigns in Vietnam, particularly aimed at younger parents from lower socioeconomic backgrounds in rural areas, to raise awareness of the harmful effects of physical violence on adolescent and child health. Policymakers and nonprofit organizations should also prioritize the implementation of culturally adapted parent education and advocacy programs to provide parents with nonviolent strategies for disciplining their children.

Adolescents who reside in the same household as a family member who has gone to prison before are more likely to be depressed. Attachment theory states that the presence of a caregiver is key to a child's sense of security, which is the foundation of the child's cognitive, emotional and social development (Bowlby, 1979). When a parent or family member is imprisoned, adolescents often experience emotional distress due to the unexpected loss of a caregiver and breadwinner, which constitutes a traumatic and adverse childhood experience (Martin, 2017). This disruption, particularly when the family member is lost to penal custody, can lead to an ambiguous loss, as children and adolescents may not fully understand the circumstances of the incarceration. This lack of clarity can result in feelings of confusion, abandonment and depression (Betz & Thorngren, 2006). A systematic review found that adolescents affected by parental incarceration often exhibit internalizing responses, such as anxiety and depression, alongside externalizing behaviors, such as substance abuse and aggression (Luk et al., 2023). In the context of Vietnamese culture, where children of incarcerated parents face significant stigma and are considered taboo, these adolescents may experience increased shame, isolation and a reluctance to seek mental health support due to societal pressures (Vang, 2021). Community-based mentoring programs, which foster trusting and supportive relationships between adolescents and other adult figures, have shown promise in mitigating the negative effects of disrupted attachment caused by family incarceration (Poehlmann-Tynan & Eddy, 2019; Stump et al., 2018). Furthermore, public awareness campaigns aimed at reducing the stigma surrounding incarcerated individuals and their families could play a crucial role in helping affected adolescents reintegrate into their communities. Given the intersection between parental incarceration and other forms of inequality, future research should explore the impact of parental incarceration on adolescents within disadvantaged and vulnerable populations in Vietnam.

This study found that adolescents who did not experience school bullying were significantly less likely to be depressed. Past systematic reviews support this finding and report that bullying can result in a 2.77-fold increased risk of depression in adolescents who were bullied as compared to those who were not bullied, a 1.73-fold increased risk of depression in adolescents who bullied others as compared to those who experienced bullying, and a 3.19-fold increased risk of depression in adolescents who bully and were bullied as compared to those who experienced neither ends (Halliday et al., 2021; Ye et al., 2023). Adolescents who are victims of bullying may have negative self-perceptions and poor self-esteem, and feel intense feelings of shame and inadequacy, resulting in their hesitation to seek external help (Duan et al., 2020). Adolescents who bully others are also at risk of depression as their aggressive and violent behaviors stem from self-loathing and depression, and further isolate themselves from their peers when they bully others (Duan et al., 2020). Past local studies have also reported an increased risk of depression, psychological distress and suicidal ideation among adolescents who were victims, perpetrators

or both of bullying across Vietnam (Le, Dunne, et al., 2017; Le, Nguyen, et al., 2017; Nguyen et al., 2019). Interestingly, one of the studies noted a difference in the nature and psychological effect of bullying between male and female Vietnamese adolescents (Le, Nguyen, et al., 2017). Policymakers, school principals and school teachers should aim to create a positive and supportive environment at school through teacher training programs to intervene against bullying at school, the introduction of an anonymous self-report channel by students of bullying incidents in school and regular small-group meetings between teachers and students (Baraldsnes, 2020). There is also a need for the development of targeted interventions for gender-specific adolescent victims of bullying due to the different nature of the violence experienced. Mental health professionals could also facilitate psychoeducational workshops and counseling services in schools to instill empathy, culture and mutual respect among adolescents (Yosep et al., 2023).

This study found that adolescents who did not experience TFSV were less likely to be depressed. A review of 16 articles reported that ~8.8% of adolescents and adults have had their sexts (texts with sexually explicit content) shared without consent, 7.2% of them have been threatened with sext distribution and 17.6% have had their images taken without their consent (Patel & Roesch, 2020). The victims of such technology-facilitated sexual violence experienced significant mental health impacts, such as lowered self-esteem, anxiety, depression and trauma (Patel & Roesch, 2020). Digital or technology-facilitated sexual violence is a gendered phenomenon – evident from most victims of digital sexual violence being female adolescents, and most perpetrators of digital sexual violence being male adolescents (Schulz et al., 2015). Sexual minority adolescents were also two to six times more likely to be victims of technology-facilitated sexual violence as compared to their heterosexual peers in America and Sweden (Patchin & Hinduja, 2018; Priebe & Svedin, 2012). Their vulnerability could be attributed to an overall higher risk of sexual victimization, lack of support from family and friends and discrimination (Gámez-Guadix & Incera, 2021; Turner et al., 2024). Schools, community organizations and health workers should also provide a nonjudgmental avenue for victims of technology-facilitated sexual violence to seek mental health support and provide online educational resources to adolescents on how they can protect themselves against digital sexual violence (Patterson et al., 2022). For long-lasting changes to prevailing societal attitudes, there is a need for the implementation of culturally adapted sexual and gender education for Vietnamese adolescents to unlearn traditional gender roles and misogyny. As this study did not individually measure the various tenets that make up digital sexual violence, future research could investigate the prevalence and risk factors of the different types of behaviors involved in digital sexual violence among Vietnamese adolescents, their differentiated consequences on their mental health and their relationship with gender and sexuality.

### Strengths and limitations

This study has a few limitations. First, the cross-sectional design restricts the ability to establish causal relationships between the identified factors and depressive symptoms among adolescents. Additionally, the use of self-reported questionnaires on sensitive topics, such as underage alcohol consumption and sexual violence, introduces potential self-reporting bias and social desirability bias. A small number of students did not complete the survey, and their responses were excluded from the analysis. Since the reasons for these

incomplete responses remain unclear, there may be unaccounted factors that could affect the generalizability of the study's findings.

Nevertheless, the study's large sample size and rigorous sampling methods contributed to the recruitment of a representative sample of the adolescent population in Can Tho City, Vietnam. Moreover, the detailed contextual description of Vietnam has enhanced the applicability of the findings and recommendations to other LMICs or Southeast Asian contexts.

### Relevance to practice

Mental health professionals should prioritize educating adolescents and their parents about the negative consequences of excessive academic pressure, exposure to physical violence at home and substance use, including smoking and alcohol consumption. Special attention should be given to adolescents at higher risk, such as those with a family history of depression or who have experienced parental incarceration, bullying or sexual violence. Psychoeducation workshops may be effective in fostering empathy and promoting a culture of mutual respect, potentially reducing rates of interpersonal violence among adolescents. Furthermore, the provision of counseling and trauma-informed care for victims of interpersonal and sexual violence could be instrumental in preventing the onset of depression in these vulnerable individuals.

School-based interventions, such as CBT and mindfulness programs, can equip adolescents with strategies to challenge distorted cognitions and build resilience against academic pressure. In addition, schools should incorporate comprehensive sexual and gender education to address and prevent sexual violence among adolescents. As schools are environments where adolescents spend a significant amount of their time, they provide an ideal setting for universal preventive mental health programs. School-based initiatives can reach a broad population of students, helping to mitigate common barriers to mental health service access, such as stigma, accessibility and cost. Creating a safe and supportive school environment, free from bullying, is essential. To achieve this, schools should implement anonymous reporting channels for bullying, facilitate small group meetings between students and teachers and ensure that teachers are trained to intervene effectively in bullying incidents. Community organizations and workers could play a key role in introducing community-based mentoring programs for highly vulnerable adolescents, including those who have experienced parental incarceration or interpersonal and sexual violence. These programs could provide a safe space for adolescents to share their experiences and receive peer support from others who have faced similar challenges. Community organizations could also be crucial for outreach and providing mental health support to adolescents who have left or dropped out of school.

Policymakers should invest additional resources into the mental health infrastructure to improve the screening, detection and treatment of depressive symptoms across a broader segment of the Vietnamese population. This investment is critical to breaking the intergenerational cycle of mental health issues and violence. Public awareness campaigns addressing the harms of corporal punishment, along with parent training programs, could help shift parents away from violent disciplinary practices. Additionally, nationwide campaigns aimed at destigmatizing and reintegrating adolescents with incarcerated relatives would further support the mental health and social reintegration of these youth.

### Conclusion

This study uncovered a high prevalence of depressive symptoms among adolescents in Can Tho City and identified several key contributing factors, including cognitive distortions, academic pressure, exposure to physical domestic violence, family incarceration, adolescent alcohol consumption and smoking, bullying and experience of technology-facilitated sexual violence. These findings underscore the urgent need for comprehensive mental health interventions that target modifiable risk factors. A coordinated, multi-level approach involving parents, mental health professionals, schools and the wider community is crucial in addressing these issues through early detection of at-risk individuals, provision of support and mental health literacy education for adolescents and key stakeholders. Additionally, equipping adolescents with skills to reframe negative thoughts, manage academic stress and protect themselves from interpersonal and sexual violence is essential for reducing the incidence of depressive symptoms in this population. Future research should further investigate the factors influencing depressive symptoms in marginalized and vulnerable groups in Vietnam, including ethnic, gender and sexual minorities, as well as low-income and rural populations.

**Open peer review.** To view the open peer review materials for this article, please visit http://doi.org/10.1017/gmh.2025.10096.

**Supplementary material.** The supplementary material for this article can be found at http://doi.org/10.1017/gmh.2025.10096.

**Data availability statement.** The data can be made available via email upon reasonable request from the corresponding author.

**Acknowledgments.** The authors would like to acknowledge that the Can Tho University of Medicine and Pharmacy, Mahidol University, the National University of Singapore, the Can Tho Department of Education and Training, other medical students, high school administrators and teachers have supported us in this study, as well as high school students and parents for participating in the study.

**Author contribution.** Nine authors contributed to the completion of this manuscript. The contributor roles for this article are as follows: TLCT: Data curation, formal analysis, investigation and manuscript writing (original draft). CR: Conceptualization, investigation, methodology, project administration, supervision and manuscript writing (original draft, review and editing). OL: Conceptualization, investigation, methodology and supervision. TTS: Conceptualization, investigation, methodology and supervision. RD: Formal analysis, investigation, validation, visualization, and manuscript writing (review and editing). SR: Validation, visualization and manuscript writing (review and editing). TTN: Validation, visualization and manuscript writing (review and editing). TTBS: Visualization and manuscript writing (review and editing). SS: Supervision, validation, visualization and manuscript writing (original draft, review and editing). All the authors reviewed the results and analysis and approved the final version of the manuscript. All authors have contributed significantly and met the authorship criteria according to the latest guidelines of the International Committee of Medical Journal Editors.

**Financial support.** This research received no specific grant from any funding agency, commercial or not-for-profit sectors.

**Competing interests.** The authors declare none.

**Ethics statement.** This study received ethical approval from Mahidol University (Certificate of Approval No. MU-CIRB 2024/153.0606), Can Tho University of Medicine and Pharmacy (No. 24.001/PCT-HDDD) and the Can Tho Department of Education and Training (No. 1518/SGDDT-CTTT) for data collection.

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
