## [Reviewer Report]

First of all, thank you for the invitation to revise the manuscript: Prevalence of Depression and Its Determinants Among Adolescents in Can Tho City, Vietnam: A Cross-Sectional Study. The main objective of this study was to investigate prevalence and factors associates with depressive symptomatology in adolescents.

The article is well written, clear and the theme is relevant specifically in LMIC’s. There are some minor changes that should be made before publishing. In what follows, there are some suggestions for improving this manuscript:

1.Title

The authors evaluated “depressive symptoms” and not “depression diagnosis”. I suggest to change Depression for Depressive symptoms, for instance.

2. Abstract

The authors wrote 1054 adolescents, but the final sample was 930, wasn’t?.

3. Introduction:

I missed I missed a sentence explaining why your article is innovative.

4. Methods

Ok

5. Results

A figure with a theoretical model would be interesting for explain better the choice of the variables.

6. Discussion

As mentioned in the title, in some parts of the discussion section the authors refer to ‘depression’ when discussing the results, but in fact, the outcome was “depressive symptoms”, not a clinical diagnosis of depression. Could the authors clarify this distinction?"

Additionally, the discussion section is overly lengthy and could be more concise and focused

---

## [Reviewer Report]

Reviewer

I have the following comments for the authors to address.

Introduction section:

1. The period of adolescence spans from 13 to 19 years of age. The rationale for selecting adolescents aged 15-18 years as the population for this study should be explained.

2. The study adopted the social-ecological model as a conceptual framework for this study. It means that the researchers derive concepts or variables from each level of the multifaceted model that tend to affect depressive symptoms in adolescents. The selected variables should indicate that each variable originates from a specific level (individual, interpersonal, community, or societal) within the model. Along with presenting the reasons for selecting variables.

3. The independent variables used to analyze for association with depressive symptoms were many. The researchers should select only independent variables that have evidence to support them and have strong measurements that accurately represent the variable.

Methodology section.

4. In the part of data analysis, page 4: A total of 29 variables with statistically significant association with depressive symptoms in univariate analyses were then selected to be entered into the multivariable logistic regression model. I have a question regarding the level of statistical significance chosen for the next step of analysis. Another question in this part is that the researchers use a multivariable logistic regression model, but in Table 3, they use a multiple regression model to examine the factors associated with the depression score. Please reconsider the statistics you use for data analysis.

5. On Table 1: What is the meaning of “Screening time per day”?

6. Part of measurement: The authors used so many variables, but you describe only 5 measures for five variables; the other variables were not present. How were they measured?

Discussion

7. Line 5, Adolescents who had more cognitive distortions felt greater academic pressure, experienced domestic physical abuse………………..to have greater rates of depressive symptoms. It appears that the author has expressed their own opinion, which is not based on previous studies or existing knowledge. Please be cautious when discussing the results, as they are often presented in this manner without proper citation.

8. Discussion of the finding, it would be better if the authors connected the results to the socio-ecological model that is the conceptual framework of this study. What variables work as individual factors, interpersonal factors, community factors, and societal factors? Please, present according to the multi-level and multi-dimensional approach that the authors summarize in the conclusion section. In addition, the factors most important in affecting depressive symptoms should be discussed first, followed by those with lower scores that also affect depressive symptoms.

---

## [Reviewer Report]

The manuscript reports findings from a study estimating the prevalence of depressive symptoms among adolescents in Can Tho City, using a sample of 1,054 students. The paper also examines various factors associated with depressive symptoms. The topic is important and relevant; however, several points require clarification and revision.

1. Sample Size Calculation

The manuscript mentions an effect size of 1.5–2 in the sample size calculation. Given that the primary objective is to estimate prevalence, please clarify why the power calculation was based on an effect size, and explain the rationale for using a value of 1.5–2.

2. Sampling and Weighting

Were the results weighted to account for the sampling strategy used by the researchers? If not, please justify why weighting was not necessary.

3. Missing Data

The authors conducted a complete case analysis, but no statistical treatment of missing data was described. Please explain why complete case analysis was chosen and whether the missing data mechanism (e.g., missing completely at random) justifies this approach.

3. Terminology

Please use the term multivariable instead of multivariate, as the regression models included only one dependent variable.

4. Statistical Reporting

When reporting point estimates along with confidence intervals, p-values do not provide additional information. Consider reporting either the confidence intervals or the p-values—but not both—for clarity and conciseness.

5. Comparison of Prevalence Estimates

In the discussion section, please clarify whether the prevalence estimates cited from Thailand, Indonesia, Myanmar, and Laos used the same cutpoint for depressive symptoms as was used in the current study.

6. Discussion of Associated Factors

The discussion of the eight factors associated with depression could be shortened and better organized. Consider structuring this section using the social-ecological framework introduced earlier in the paper.

---

## [Editor Report]

Dear Dr Rattanapan

Your manuscript ‘Prevalence of Depression and Its Determinants Among Adolescents in Can Tho City, Vietnam: A Cross-Sectional Study’ has now been reviewed,

---

## [Reviewer Report]

Please consider the following in revising your manuscript:

- The manuscript continues to mention effect size without clearly explaining its meaning or relevance. The citation provided refers to design effect, which is a different concept. The authors need to clarify whether they intended to refer to effect size or design effect, and ensure that the correct concept is cited and accurately described in the manuscript.

- In their response to reviewers, the authors mentioned the missing data rate and their assessment of potential bias. This information should be included in the manuscript itself so that readers can see the extent of missing data and the steps taken to ensure it does not bias the results.

- Not applying sampling weights in population-based surveys can lead to distorted results. Although the rationale for not applying weights was provided in the authors’ response to reviewers, the manuscript should explicitly include this explanation so that readers understand the analytic choices.

---

## [Reviewer Report]

I really appreciated the revised manuscript. However, I have a few recommendations in the introduction section. The authors have explained the reasons for selecting the expected determinants of depressive symptoms among adolescents based on existing knowledge and previous studies. The content was a response to a reviewer; however, readers may not understand why the variables were chosen. It would be beneficial if the authors clearly defined the source (in-text citation) and the idea of the selected original independent variables in the introduction with concise statements. Additionally, please categorize the variables by level, specifically individual, interpersonal, and community.

---

## [Editor Report]

Dear Author,

Your revised manuscript: ‘Prevalence of Depressive Symptoms and Their Determinants Among Adolescents in Can Tho City, Vietnam: A Cross-Sectional Study’, has now been reviewed,

---

## [Reviewer Report]

I acknowledge that the author has revised the manuscript. Thank you for inviting me to be a reviewer.

---

## [Editor Report]

Dear Prof. Rattanapan,

Your revised manuscript “Prevalence of Depressive Symptoms and Their Determinants Among Adolescents in Can Tho City, Vietnam: A Cross-Sectional Study”,has now been reviewed,